# *Tamarindus indica* Extract as a Promising Antimicrobial and Antivirulence Therapy

**DOI:** 10.3390/antibiotics12030464

**Published:** 2023-02-24

**Authors:** Mohamed F. Ghaly, Marzough Aziz Albalawi, Mahmoud M. Bendary, Ahmed Shahin, Mohamed A. Shaheen, Abeer F. Abu Eleneen, Mohammed M. Ghoneim, Ayman Abo Elmaaty, Mohamed F. M. Elrefai, Sawsan A. Zaitone, Amira I. Abousaty

**Affiliations:** 1Botany and Microbiology Department, Faculty of Science, Zagazig University, Zagazig 44519, Egypt; 2Department of Chemistry, Alwajh College, University of Tabuk, Tabuk 71491, Saudi Arabia; 3Department of Microbiology and Immunology, Faculty of Pharmacy, Port Said University, Port Said 42526, Egypt; 4Microbiology and Immunology Department, Faculty of Medicine, Zagazig University, Zagazig 44519, Egypt; 5Clinical Pathology Department, Faculty of Medicine, Al-Azhar University, Cairo 11311, Egypt; 6Department of Pharmacy Practice, College of Pharmacy, AlMaarefa University, Ad Diriyah 13713, Saudi Arabia; 7Medicinal Chemistry Department, Faculty of Pharmacy, Port Said University, Port Said 42526, Egypt; 8Anatomy, Histology, Physiology and Biochemistry Department, Faculty of Medicine, Hashemite University, Zarqa 13116, Jordan; 9Anatomy and Embryology Department, Faculty of Medicine, Ain Shams University, Cairo 11566, Egypt; 10Department of Pharmacology and Toxicology, Faculty of Pharmacy, University of Tabuk, Tabuk 71491, Saudi Arabia; 11Department of Pharmacology and Toxicology, Faculty of Pharmacy, Suez Canal University, Ismailia 41522, Egypt

**Keywords:** *Tamarindus indica*, MIC, MBC, FICI, antivirulence, molecular docking

## Abstract

The worldwide crises from multi-drug-resistant (MDR) bacterial infections are pushing us to search for new alternative therapies. The renewed interest in medicinal plants has gained the attention of our research group. *Tamarindus indica* L. (*T. indica*) is one of the traditional medicines used for a wide range of diseases. Therefore, we evaluated the antimicrobial activities of ethanolic extract of *T. indica.* The inhibitions zones, minimum inhibitory concentration (MIC), minimum bactericidal concentration (MBC), and fractional inhibitor concentration indices (FICI) against Gram+ve and −ve pathogens were detected. The bioactive compounds from *T. indica* extract were identified by mass spectroscopy, thin-layer chromatography, and bio-autographic assay. We performed scanning electron microscopy (SEM) and molecular docking studies to confirm possible mechanisms of actions and antivirulence activities, respectively. We found more promising antimicrobial activities against MDR pathogens with MIC and MBC values for *Staphylococcus aureus* (*S. aureus*) and *Pseudomonas aeruginosa* (*P. aeruginosa*), i.e., (0.78, 3.12 mg/mL) and (1.56, 3.12 mg/mL), respectively. The antimicrobial activities of this extract were attributed to its capability to impair cell membrane permeability, inducing bacterial cell lysis, which was confirmed by the morphological changes observed under SEM. The synergistic interactions between this extract and commonly used antibiotics were confirmed (FICI values < 0.5). The bioactive compounds of this extract were bis (2-ethylhexyl)phthalate, phenol, 2,4-bis(1,1-dimethylethyl), 1,2-benzenedicarboxylic acid, and bis(8-methylnonyl) ester. Additionally, this extract showed antivirulence activities, especially against the *S. aureus* protease and *P. aeruginosa* elastase. In conclusion, we hope that pharmaceutical companies can utilize our findings to produce a new formulation of *T. indica* ethanolic extract with other antibiotics.

## 1. Introduction

Recently, the failure to manage and treat infectious diseases associated with resistant bacteria are well-established globally. Further, increased attention is required from all healthy organizations throughout the world to establish promising strategies and funding for researchers to avoid the proliferation of this issue. The control of microbial infections started with the discovery and development of antibiotics. Antimicrobial drugs are critical for lowering the global burden of infectious illnesses [1]. On the other hand, antibiotic resistance has become more frequent worldwide, often due to new resistance mechanisms, which allow the increase and spreading of resistant bacterial strains [2]. Due to the scarcity of antibiotics effective in controlling resistant pathogenic bacteria, the spread of multi-drug-resistant (MDR) strains has become a public health concern [3].

Virulence factors have several essential roles in the increasing mortality and morbidity rates, leading to life threating diseases [4]: allowing microbial breakthroughs the host defense power; increasing the rate of evolutionary processes; and increasing the pathogenesis of the microbial cells. Treatment failures are being recorded at an unprecedented rate, especially for the infections with multi-drug-resistant (MDR) and multi-virulence pathogens [5]. The treatment failures continue to rise due to the transmissibility of genetic components encoding both virulence and antibiotic resistance, in addition to the fact that the discovery and development of new antimicrobial drugs has begun to dry up. Therefore, new trends in the development of antimicrobial strategies include the inhibition of microbial virulence arrays rather than growth pathways. 

Accordingly, the combination of antivirulence compounds with other antimicrobial drugs can be used to avoid treatment failures. Several antivirulence mechanisms have previously been reported, including toxin neutralization, inhibition of biofilm formations, inhibition of cell adherence, downregulating virulence gene expression, and binding with virulence proteins and enzymes [6,7]. Several new chemical compounds have been found to have antivirulence activities; meanwhile, their potential side effects on different systems in the human body remain the greatest clinical challenge impeding the use of these compounds. Therefore, we return again to the use of natural compounds from medicinal plants to fight the virulence factors of resistant pathogens. Medicinal plants have many advantages, such as their safety and several antimicrobial mechanisms. Several secondary metabolites have been found inside medicinal plants which can be used to treat different diseases [8]. Identification of the bioactive compounds of medicinal plants is essential to select the most active antimicrobial compounds. Several methods can be used to identify the biologically active compounds with high separation efficiency, such as bio-autography assay and layer chromatography (TLC) [9]. 

Plant phytochemicals, such as flavonoids, phenolic compounds, alkaloids, and tannins, generate secondary metabolisms and exhibited antimicrobial effects against human pathogens and phytopathogens in plants [1,2]. One of the most important multipurpose medicinal plants is the Tamarind tree, which was first produced in India and used in treatment of GIT disorders, such as dysentery and other diarrhea-related illnesses; the healing of wounds; diabetes; hepatic disorders; and some types of helminthic and bacterial infections thanks to the high levels of phenolic compounds, cardiac glycosides, crude proteins, and carbohydrates. In this respect, *Tamarindus indica* L (*T. indica*) has significant medicinal properties, as reported in several phytochemical studies [10]. The *T. Indica* pulps are used for many industrial purposes, such as the production of flavoring agents and sweet meats; meanwhile, the seeds are used in food manufacturing to improve texture and viscosity. In the same context, salads, stews, and soups are made from the leaves and flowers of *T. Indica* in several areas throughout the worlds. The seeds of *T. indica* contain many active biological compounds such as fatty acids (palmitic acid, eicosanoic acid), phenolic antioxidants, campesterol, and b-amyrin [11]. On the other hand, leaf extracts of *T. indica* are known to contain many compounds, such as flavonoids (e.g., naringenin, epicatechin, catechin, and apigenin), polyphenols, β-carotene, and ascorbic acid [12]. These phenolic molecules, with various chemical structures, could be exploited for therapeutic intervention thanks to their effective biological activity [13,14]. Furthermore, the bioactivities of these metabolites are linked to polyphenol interactions with biomolecules, such as carbohydrates, lipids, and proteins, which cause cell permeability alterations in target bacteria and, finally, disruption their cell walls [14]. The goal of this investigation was to assess the antibacterial and antivirulence potentials of a ethanolic extract from fruits of *T. indica* against multi-drug-resistant (MDR) bacterial strains, such as *Staphylococcus aureus* (*S. aureus*) and *Pseudomonas aeruginosa* (*P. aeruginosa*), isolated from Egyptian hospital environments. 

## 2. Results

### 2.1. Antibacterial Effects of T. indica against MDR Bacteria

The antimicrobial activities of the ethanolic extract from *T. indica* against both MDR Gram+ve (*S. aureus)* and Gram−ve *(P. aeruginosa)* isolates were announced in our study. The inhibition zone diameters of *T. indica* ethanolic extract were 31 ± 0.17 and 20 ± 0.21 mm for *S. aureus* and *P. aeruginosa*, respectively. The promising results of inhibition zones diameters were matched with both MIC and MBC values. Interestingly, the MICs and MBCs of *T. indica* ethanolic extract were 0.78 ± 0, 3.12 ± 0 mg/mL for *S. aureus* and 1.56 ± 0, 3.12 ± 0 mg/mL for *P. aeruginosa*, respectively. Therefore, the antimicrobial activity of this extract on the tested Gram+ve bacteria was higher than on the tested Gram−ve bacteria. Furthermore, the interactions between the tested extract and the commonly used antibiotics (imipenem, amikacin, and ofloxacin) were detected. Interestingly, synergistic interactions were recorded for these combinations, with FICI values < 0.5. Therefore, the antimicrobial activities of the tested antibiotics were increased in the presence of *T. indica* ethanolic extract.

### 2.2. The Possible Antimicrobial Mechanisms of Actions

The leakage of K^+^ ions in response to the *T. indica* ethanolic extract at MIC concentration was measured by an atomic absorption spectrophotometer. *T. indica* ethanolic extract caused a rapid increase in ion leakage in both *S. aureus* (Gram+ve) (Figure 1A) and *P. aeruginosa* (Gram−ve) (Figure 1B) during the first 60 min after exposure. This was still increasing after 60 min, but at a low rate. The nucleotide leakage in both *S. aureus* (Figure 1C) and *P. aeruginosa* (Figure 1D) gradually increased upon treatment with the MIC concentration. Thus, Gram+ve and Gram−ve bacteria possess a strong sensitivity to *T. indica* extract, which causes great cell membrane damage.

Moreover, SEM was performed to observe the morphological effects of *T. indica* on the MDR *P. aeruginosa* and *S. aureus* isolates. In contrast to untreated *P. aeruginosa* cells, which displayed typical bacilliform with uniformity in size and distribution (Figure 2A), cells treated with the extract had irregular, withered, coarse surfaces; lysis of cell membranes; and leakage of cellular contents, forming aggregations and adhesions (Figure 2B). Similar alterations were observed in cells of *S. aureus* treated with the same extract (Figure 2C,D). 

### 2.3. Phytochemical Analysis of the Bioactive Molecules

Phytochemical analysis revealed that *T. indica* ethanolic extract includes phenols, flavonoids, alkaloids, quinones, tannins, saponins, and terpenoids. Paper TLC was used to purify active antibacterial compounds, which was followed by a bio-autography assay. The dried fraction of *T. indica* ethanolic extract with an Rf value of 0.4, among all other ethanolic extract fractions, demonstrated antibacterial activity against *P. aeruginosa* and *S. aureus*. This fraction was subjected to UV, IR, and MS scanning. From the UV profile, two major peaks were evidenced at λ = 264.5 and 214.5 nm (Figure 3A). In addition, the IR spectrum (Figure 3B) showed principal peaks at 3351, 2934, 1736, 1633, 1246, and 1076 cm^−1^, corresponding to OH, aliphatic C-H, C=O, C=C (aromatic ring), O=C-O-, and aromatic C-H bands. MS of the extract revealed three major compounds: Bis (2-ethylhexyl) phthalate (C_24_H_38_O_4_) with an MW of 390,277 (Figure 3C); aromatic hydrocarbon [phenol, 2,4-bis (1,1-dimethylethyl)] (C_14_H_22_O) with an MW of 206.167 (Figure 3D), and 1,2-benzenedicarboxylic acid, bis(8-methylnonyl) ester (C_28_H_46_O_4_) with an MW of 446.339 (Figure 3E).

### 2.4. GC-MC Analysis

According to mass spectroscopy analysis, the chemical composition of purified ethanolic extract from *T. indica* contained the following molecules: (i) bis (2-ethylhexyl) phthalate (C_24_H_38_O_4_) with an MW of 390, 277 (56.5%), as illustrated in Figure 4; (ii) the aromatic hydrocarbon (phenol, 2,4-bis (1,1-dimethylethyl)-(C_14_H_22_O)) with an MW of 206.16 (43.6%), as illustrated in Figure 4; (iii) 1,2-benzenedicarboxylic acid, bis (8-methylnonyl) ester (C_28_H_46_O_4_) with an MW of 446.339 (22.4%).

### 2.5. Molecular Docking Studies

From all the virulence proteins expressed by *S. aureus* and *P. aeruginosa* which were tested by the molecular docking studies, only the *S. aureus* protease and *P. aeruginosa* elastase could bind with bioactive compounds of *T. indica.* Regarding *S. aureus*, it was discovered that the investigated compounds (Bis (2-ethylhexyl) phthalate (1), phenol, 2,4-bis (1,1-dimethylethyl) (2), and 1,2-benzenedicarboxylic acid, bis(8-methylnonyl) ester (3) could interact with the *S. aureus* protease through pi-H bond with PRO153, at binding scores of −6.52, −4.46, and −7.36 Kcal/mol and with RMSD values of 1.74, 1.52, and 1.58 Å, respectively. It is worth noting that the benzene ring was responsible for pi-H bond formation with PRO153 for the three investigated compounds, as shown in Table 1 and Figure 5. However, regarding *P. aeruginosa*, it was shown that compound 1 could interact with *P. aeruginosa* elastase at a binding score of −7.47 Kcal/mol, and the value of RMSD was 2.38 Å. Obviously, the carboxylate group at the phthalate moiety in compound 1 had the ability to form H-bond with TYR155 at 2.85 Å. The terminal side chain of compound 1 could form two H-pi bonds with HIS140, at distances of 3.70 and 4.40 Å. In addition, compound 2 exhibited a binding score of −4.74 Kcal/mol, with an RMSD value of 0.98 Å. Notably, the *tert*-butyl moiety in compound 2 was able to form an H-pi bond with HIS140 at a distance of 4.25 Å. Furthermore, compound 3 displayed a binding score equal to −7.45 Kcal/mol and an RMSD value equal to 1.90 Å. Notably, the phenyl ring of compound 3 was able to form a pi-H bond with ASN112 at a distance of 4.79 Å. The terminal side chain of compound 3 was able to form an H-pi bond with HIS140 at a distance of 4.43 Å. Compounds 1, 2, and 3 were able to form metal bonds with GLU141 at a distance of 1.72 Å, as illustrated in Table 1 and Figure 6.

## 3. Discussion

The wide spread of MDR bacterial and fungal pathogens has created several health problems [15,16], especially throughout those countries that did not follow up on the infection control guidance. The discovery and development of antimicrobial therapies are normally conducted with very slow steps, which is not in line with the rate of evolution of antimicrobial resistance mechanisms to commonly used antibiotics. In the same context, the therapeutic switching of already-used medicine [17] and the renewed interest in medicinal plants [18] may compensate for the wide gap in solutions for this issue. Therefore, the use of complementary and alternative medicines, especially natural compounds and essential oils, with certain precautions, are the perfect choice to prevent the compounding of this crisis. The use of medicinal plants must occur under full medical supervision, without any self-medication, to avoid drug interactions, in addition to other adverse effects [19]. 

*T. indica* extracts from various plant parts have been used for several therapeutic purposes [20]. In this study, *T. indica* ethanolic extract was selected, and its antimicrobial and antivirulence activities were evaluated against Gram+ve and Gram−ve resistant pathogens. Generally, a broad spectrum of antibacterial activity with low MIC and MBC values was observed for the tested ethanolic extract compared to ordinary aqueous extract [20]. In this study, the promising use of ethanolic extract from *T. indica* as an alternative and complementary therapy for resistant pathogens was confirmed by the large zones of inhibitions. Additionally, the MIC and MBC values were detected for *S. aureus* (0.78, 3.12 mg/mL) and *P. aeruginosa* (1.56, 3.12 mg/mL), respectively. Parallel to our finding, several authors reported the antimicrobial activities of ethanolic extract of *T. indica* through MIC and MBC values, thus confirming our hypothesis [20,21,22]. Therefore, the success of our postulates regarding the antimicrobial activities of *T. indica* makes reconsidering the use of other medicinal plants an urgent necessity.

In fact, the in vitro antimicrobial potential of these natural compounds did not reflect the overall bacterial response in vivo, since it was tested in broth rather than in a physiological human body, in addition to the bioavailability problems [23]. Therefore, we cannot suggest the use of *T. indica* extract as the sole drug for treating resistant pathogens. In the same context, resistance to commonly used antibiotics such as imipenem, amikacin, and ofloxacin were previously reported. For that, we suggest the use of a combination of any of these antibiotics and *T. indica* extract. Synergistic interactions between these combinations were detected (FICI > 0.5) against both Gram+ve and Gram−ve bacteria. Confirming our finding, the co-admixing of antibiotics with natural compounds and/or essential oils had huge success in treating MDR bacterial and fungal infections, in contrast to the use of each one alone [24,25,26]. Furthermore, the use of medicinal plants can reduce the duration of use, dose, and toxicity hazards associated with antibiotics, and decrease the possibility of the emergence of new resistant strains [27].

It has also been reported that a huge number of bioactive compounds were found in various parts of medicinal plants. These bioactive compounds were diverse in their chemical structure, and their concentration was not the same in each part of the medicinal plants. Therefore, it is essential to determine the exact bioactive compounds of medicinal plant extract. In this report, GC-mass and other spectrophotometer analyses of the ethanolic extract of *T. indica* revealed several phytochemicals, including phenolic content, flavonoids, alkaloids, quinones, tannins, saponins, and terpenoids. This finding was in agreement with other *T. indica*-related phytochemical studies [28]. The antimicrobial activity of this extract may be attributable to its phenolic compounds [28]. Chemical analysis of the extract revealed three major compounds: (i) Bis (2-ethylhexyl) phthalate (DEHP, C_24_H_38_O_4_); (ii) aromatic hydrocarbon [phenol, 2,4-bis (1,1-dimethylethyl)] (PD, C_14_H_22_O); and (iii) 1,2-benzenedicarboxylic acid, bis(8-methylnonyl) ester (C_28_H_46_O_4_). The DEHP, a major bioactive compound in this extract, showed a broad spectrum of antibacterial activity against both G+ve and G−ve bacteria compared to other secondary metabolites [29]. The amount of these phenolic compounds, which are present in almost every part of this medicinal plant, varied according to the extraction method, geographical location, and climatic conditions [30]. 

The novelty of this study is the determination of antimicrobial mechanisms by various methods, in addition to the assessment of the antivirulence activity of *T. indica* extract by molecular docking. In this study, the microbial cytoplasmic membrane is the main target site of the bioactive compounds of this extract. Similar studies documented the mechanism of action via inhibition of protein and DNA synthesis, increasing cell membrane and wall permeability as well as lysing the cells [31,32]. The results obtained in this study revealed that *T. indica* ethanolic extract caused a rapid increase in ion leakage, especially of K^+^ ions, and nucleotides in both *S. aureus* (Gram+ve) and *P. aeruginosa* (Gram−ve); this was confirmed by an atomic absorption spectrophotometer. Additionally, the treated isolates showed irregular, withered, and coarse surfaces; lysis of the cell membrane; and leakage of cellular contents, forming aggregations and adhesions under SEM. Parallel to our findings, it was confirmed that the phenolic compounds acted on the bacterial cytoplasmic membrane as the essential intercellular materials of the treated pathogens, such as nucleic acids and other ions, released into the extracellular solution by cellular leakage [33], and similar observations have been documented by other studies [31]. Furthermore, the antivirulence activities of the *T. indica* ethanolic extract were assessed by a molecular docking study. All of the bioactive compounds showed good binding capacities with the *S. aureus* protease and *P. aeruginosa* elastase. The measurement of binding scores, RMSD values, and amino acid interactions of the investigated compounds of the tested extract with the *S. aureus* proteases and *P. aeruginosa* elastase confirmed these antivirulence activities. The DEHP affected the intercellular communication in the bacteria and resulted in a significant reduction in biofilm, extracellular polysaccharide, prodigiosin, lipase, haemolysin, and protease, thus increasing the susceptibility of bacteria to conventional antibiotics when administered synergistically [29]. In addition, 1,2-benzenedicarboxylic acid, bis (8-methylnonyl) ester is one of putative compounds found in many plants, and is known to have antimicrobial activity [34].

## 4. Materials and Methods

### 4.1. Microorganisms, Plant Materials, and Extraction

The MDR bacterial isolates which were used in this study, such as *S. aureus* ATCC25923 and *Pseudomonas aeuginosa* ATCC 27853 were kindly provided from the microbiological units of Zagazig University Hospitals. Furthermore, these isolates were confirmed by molecular detection of specific *16S RNA* genes using the previously described primers. Additionally, the MDR patterns for these isolates were confirmed by the Kirby Bauer Disc Diffusion Method according to CLSI, 2020 [35]. 

Tamarind (*Tamarindus indica* L.) is a medicinal plant used commonly in Egyptian folk medicine. Fresh fruit of *T. indica*, which was planted in Southern Egypt (Aswan city), was purchased from a local supplier in Zagazig City, Egypt. This fresh fruit was used to prepare the antimicrobial ethanolic extract used in the investigation as follows. A 50-g sample of dry fruit powder was added to 500 mL of methanol 80% and continuously shaken for 48 h at room temperature. The ethanolic solution was then centrifuged at 5000 rpm for 10 min and filtered through 1 layer of Whatman No. 1 filter paper. The supernatant was evaporated using a rotary vacuum evaporator under 34–36 kPa pressure at 45 °C. The pellet was dissolved in distilled water containing 2% dimethylsulfoxide to form stock solutions with 25 mg/mL concentration [36]. 

### 4.2. Evaluation of Antimicrobial Activities 

#### 4.2.1. Agar Diffusion Assay by Filter Paper Disc Method

The antibacterial activity of ethanolic *Tamarindus indica* extract was evaluated in triplicate by the disc diffusion method [37]. Pure bacterial isolates were sub-cultured in Muller–Hinton agar medium at 37 °C for 4 h. The density of the bacterial suspension was adjusted to 10^6^ CFU/mL, equivalent to standard barium sulfate (0.5 McFarland). Then, 3 layers of sterile filter paper discs (Whatman No. 3, 6 mm diameter) were saturated with the fruit extract, left to dry for 1 h, and then placed on the surface of the agar plate and incubated for 24 h at 37 °C. Antibacterial activity was evaluated by measuring the entire diameter of the inhibition zone in mm. 

#### 4.2.2. Estimation of Minimum Inhibitory Concentration (MIC) and Minimum Bactericidal Concentration (MBC) 

MIC is the lowest concentration that inhibits visible bacterial growth in liquid media, whereas MBC is the lowest concentration at which no growth occurs in solid media. The MICs and MBCs of the ethanolic extract from *T. indica* were determined in triplicate by the broth microdilution method [38]. To obtain the appropriate suspensions needed for each experiment, the stock solution of tamarind extract (25 mg/mL) was diluted in nutrient broth to obtain twofold serial dilutions ranging from 0.195 to 12.5 mg/mL. The bacterial broth suspensions were prepared at 10^8^ CFU/mL from overnight cultures. Each experiment used positive and negative controls, the first consisting of tubes containing a bacterial suspension and nutrient broth, and the second of tubes containing the extract from *T. indica* and nutrient broth. All tubes were incubated for 24 h at 37 °C and examined for turbidity (λ = 600 nm) to detect the MIC value. Regarding the determination of MBC, 100 µL of MIC concentration and the 2 highest concentrations were introduced onto the nutrient agar plate and incubated at 37 °C for 24 h to determine their MBC values.

#### 4.2.3. Evaluation of Co-Admixture of the Ethanolic Extract from *T. indica* with the Commonly Used Antibiotics by Checkerboard Method

The degree of interaction between the tested extract and other antibiotics (imipenem, amikacin, and ofloxacin) was assessed by determination of the fractional inhibitory concentration (FICI) values in triplicate according to [39,40]. The MIC values of the tested extract alone and in the presence of other antibiotics were detected. Additionally, the MIC values of the antibiotics alone and in the presence of tested extract were measured. The FICI values were detected according to the following equation: FICI = (MIC of tested extract in combination/MIC of tested extract alone) + (MIC of antibiotic in combination/ MIC of antibiotic alone). The synergistic and antagonistic interactions were obtained for FICI ≤ 0.5 and FICI ≥ 4, respectively. On the other hand, additive and indifference effects were expected when 0.5 < FICI ≤ 1 and 1 < FICI < 4, respectively. 

### 4.3. Assessment the Possible Antibacterial Mechanisms of T. indica Ethanolic Extract

We assessed the cell membrane integrity by measuring both the K^+^ level and nucleotide leakage. The K^+^ leakage was determined following the method reported previously [41], with minor modifications. Briefly, bacteria were allowed to grow overnight in nutrient broth in a shaking incubator at 37 °C. Then, normal saline was used to wash the cells three times, and then cells were resuspended in 1 mmol/L glycylglycine (Sigma, USA) buffer solution with a pH value of 6.8 [42]. Bacteria were treated with the studied extract at the detected MIC, and incubated in a shaking incubator at 37 °C. After that, we took the bacterial cell suspensions after 0, 10, 20, 40, 60, 80, 100, and 120 min and filtered them through a membrane (0.22 µm pore-size membrane, Sartorius, Gottingen, Germany) to remove any bacteria. We determined the K^+^ concentration in the supernatant by applying an atomic absorption spectrophotometer (900T, Perkin-Elmer Ltd., Beaconsfield, UK) at λ = 766.5 nm. In reference to previously established standard K+ solutions, the absorbance was converted to K^+^ concentration (ppm). The experiments were conducted in triplicate, and the obtained data value averages are reported herein. 

The bacterial nucleotide leakage was measured upon treatment with the studied extract [43]. After incubation of the bacterial suspensions with MIC concentrations of the extracts at 37 °C and 150 rpm, samples were taken after 1, 2, 4, 6, and 8 h and filtered through a 0.22 µm pore-size membrane for the removal of the bacterial cells. The absorbance of the filtrate was detected utilizing a UV-spectrophotometer at λ = 260 nm. The nucleotide leakage was confirmed to be a valid indicator of cytoplasmic membrane damage. 

Scanning electron microscopy (SEM) was also employed to observe morphological changes caused in *S. aureus* and *P. aeruginosa* by the ethanolic extract of *T. indica*. Cultures of tested microorganisms were treated with the detected MIC, then incubated for 6 h at 37 °C. After incubation, bacterial cells were pelleted by low-speed centrifugation (4000 rpm for 15 min); washed with sterile, distilled water; and fixed with 3% glutaraldehyde in 0.1 M phosphate buffer for 4 h at 4 °C. Then, cells were exposed to a secondary fixation with 2% aqueous solution of osmium tetroxide for 60 min at room temperature, and were then serially dehydrated with 75, 95, and 100% ethanol. The last drying step was performed over anhydrous CuSO_4_ for 15 min. After finishing the drying step, we mounted the cells on stubs of 12.5 mm diameter, attached them with sticky tabs, and then coated them in an Edwards S150B sputter coater with 25 nm thickness. Non-treated cells were used as negative controls. Small cell samples of the treated bacteria and the relative controls were examined with SEM (JEOL, Japan) at an accelerating voltage of 20 kv [44].

### 4.4. Chemical Identification of Bioactive Substances

#### 4.4.1. TLC and Bio-Autographic Assays

Thin layer chromatography (TLC) was carried out to identify the bioactive fractions of *T. indica* extract. First, its powder was dissolved in ethyl acetate and spotted by capillary tubes on TLC paper (20 × 20 cm) using running solvents chloroform/methanol (6:4, *v*/*v*). The detected fractions were then dissolved in methanol and dried. The retention factor (Rf) values of each fraction were calculated, and the antimicrobial activity of the dried fractions was re-tested against the selected pathogenic bacteria using bio-autographic assay. TLC-dried fractions were placed on the surface of a Mueller–Hinton agar plate seeded with each microbe and incubated at 37 °C for 24 h. After incubation, the clear zone that appeared on the media was taken as proof of the antibacterial efficacy of the tested extract [45,46].

#### 4.4.2. Phytochemicals Analysis

The active ingredients of the *T. indica* ethanolic extract were analyzed for the presence of different phytochemicals according to standard procedures [47]. The structure of the purified active components of *T. indica* was analyzed using data from a wide range of spectroscopic techniques, such as ultraviolet (UV), infrared (IR), and mass spectroscopy (MS), at the Regional Centre for Mycology and Biotechnology, AL-Azhar University, Cairo (Egypt). 

#### 4.4.3. GC-MS Analysis

Identification of the bioactive substances from *T. indica* was conducted at the National Research Center, Cairo, Egypt. A GC/MS-QP –1000 -Mass spectrophotometer (SHIMADU, Kyoto, Japan) instrument was used for analyses. For interpretation of the mass spectroscopy (GC-MS), we used the database of the Chemical Abstracts Service (CAS). The spectrum of unknown components was compared with the spectrum of known molecules stored in the CAS and Wiley 6 N libraries [48,49]. We recorded the retention time, molecular weight (M.Wt), molecular formula, and composition percentage in the sample material, following a previously published method [50]. 

### 4.5. Molecular Docking Study

#### 4.5.1. Molecular Docking (In Silico) Studies

A molecular docking study, which afforded us further insights into the inhibitory potential of the investigated compounds, was used in this study. The nuclei of the detected bioactive compounds of *T. indica* were evaluated against all virulence proteins expressed by the tested pathogens. The potential of the investigated compounds for the virulence proteins *S. aureus* and *P. aeruginosa* was pursued via molecular docking using an MOE 2019 suite [51].

#### 4.5.2. Preparation of the Investigated Compounds

By using PerkinElmer ChemOffice Suite 2017, the bioactive compounds of the tested extract were chemically drawn to make them ready for the molecular docking program [52,53]. We uploaded the investigated compounds to one database and saved them as an MDB extension file.

#### 4.5.3. Preparation of the Proteases of *S. aureus* and *P. aeruginosa*

All virulence proteins of the X-ray structure of *S. aureus* and *P. aeruginosa* were detected from an online protein data bank website, and downloaded with PDB entries 4INK [54] and 1EZM [55]. Accordingly, the sequence of the target protein chain was identified and protonated; then, the broken bonds were connected and fixed. Before beginning the docking process, the virulence proteins of the tested pathogens were energetically minimized [52,53].

## 5. Conclusions

This study revealed that *T. indica* ethanolic extract had a variety of in vitro antibacterial activities against MDR Gram+ve and Gram−ve isolates. It also had synergistic effects with conventional antibiotics (imipenem, amikacin, and ofloxacin) and reduced their MICs. The mechanism of activity showed that the extract was able to influence the cellular membrane permeability, as evidenced by potassium and nucleic acid leakage, resulting in cell lysis and death. Additionally, the bioactive compounds showed a good binding capacity with the *S. aureus* protease and *P. aeruginosa* elastase confirming the antivirulence activities of this extract. Therefore, we introduced a new combination of *T. indica* ethanolic extract with other antibiotics to fight MDR pathogens and to avoid treatment failure. Further studies are still needed to unequivocally determine the activity of these molecules and to evaluate their antimicrobial effects individually.

## Figures and Tables

**Figure 1 antibiotics-12-00464-f001:**
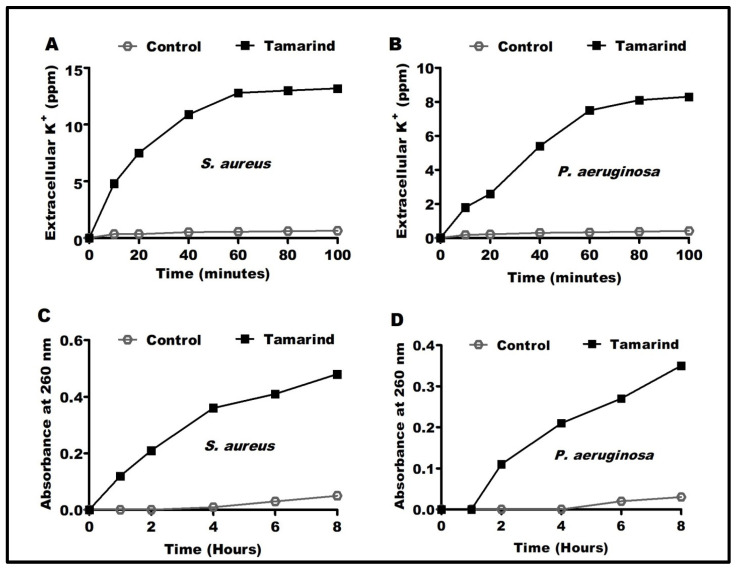
Effect of *T. indica* ethanolic extract on K+ ions released from *S. aureus* (**A**), K+ ions released from *P. aeruginosa* (**B**), nucleotides released from *S. aureus* (**C**), and nucleotides released from *P. aeruginosa* (**D**).

**Figure 2 antibiotics-12-00464-f002:**
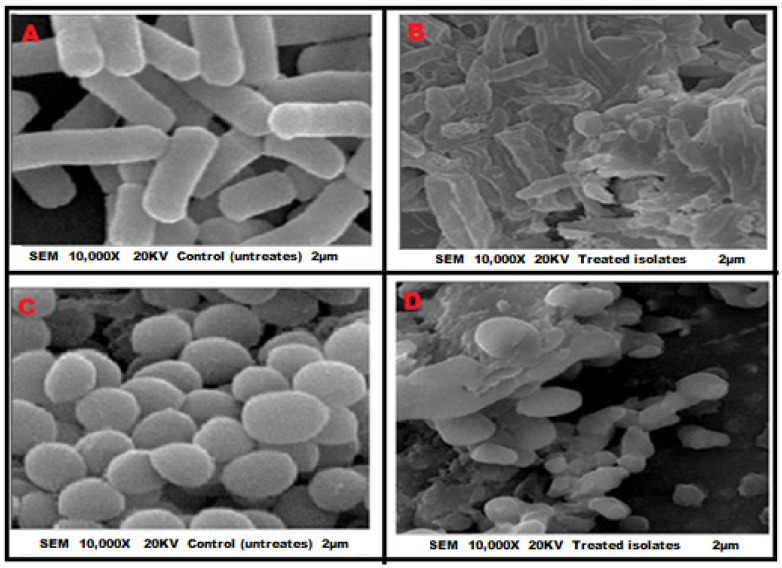
Scanning electron micrographs of (**A**) *P. aeruginosa* before, (**B**) *P. aeruginosa* after, (**C**) *S. aureus* before, and (**D**) *S. aureus* after eight hours of treatment with *T. indica* ethanolic extract at sub-MIC. Magnification: 10,000×; voltage: 20 Kv; scale: 2 µm.

**Figure 3 antibiotics-12-00464-f003:**
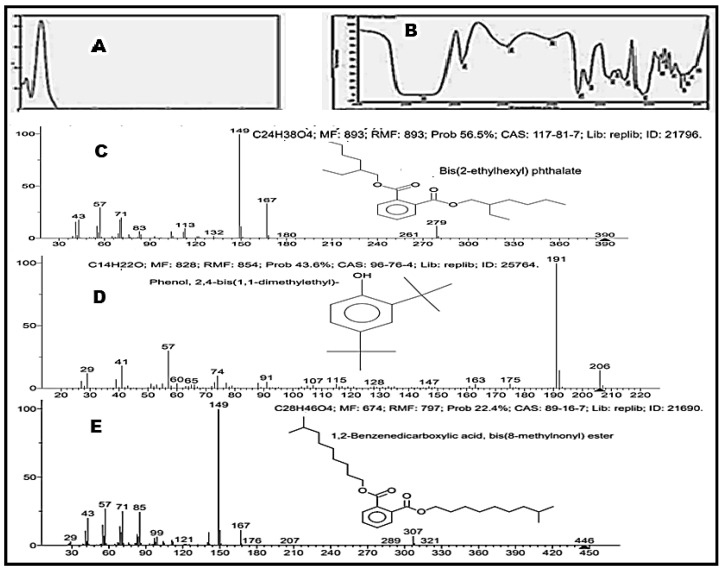
Identification of the bioactive compounds in *T. indica* ethanolic extract. (**A**) UV and (**B**) IR spectrum of the purified *T. indica* fraction with Rf 0.4. Mass spectrum revealed structures of (**C**) Bis (2-ethylhexyl) phthalate (C_24_H_38_O_4_) with a ratio of 65.5% and a molecular weight of 390,277 (**D**), phenol, 2,4-bis (1,1-dimethylethyl) (C_14_H_22_O) with a ratio of 43.6% and a molecular weight of 206.167, and (**E**) 1,2-benzenedicarboxylic acid, bis(8-methylnonyl) ester (C_28_H_46_O_4_) with a ratio of 22.4% and a molecular weight of 446.339 from *T. indica* ethanolic extract.

**Figure 4 antibiotics-12-00464-f004:**
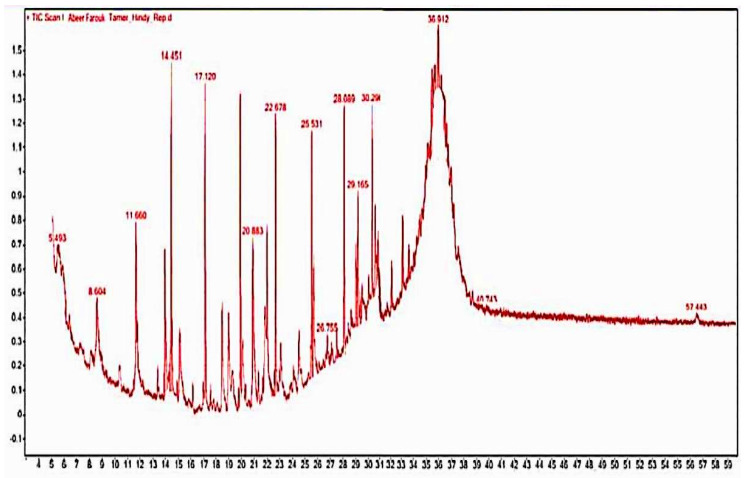
The GC−MC analysis detected the chemical composition of purified *T. indica* ethanolic extract.

**Figure 5 antibiotics-12-00464-f005:**
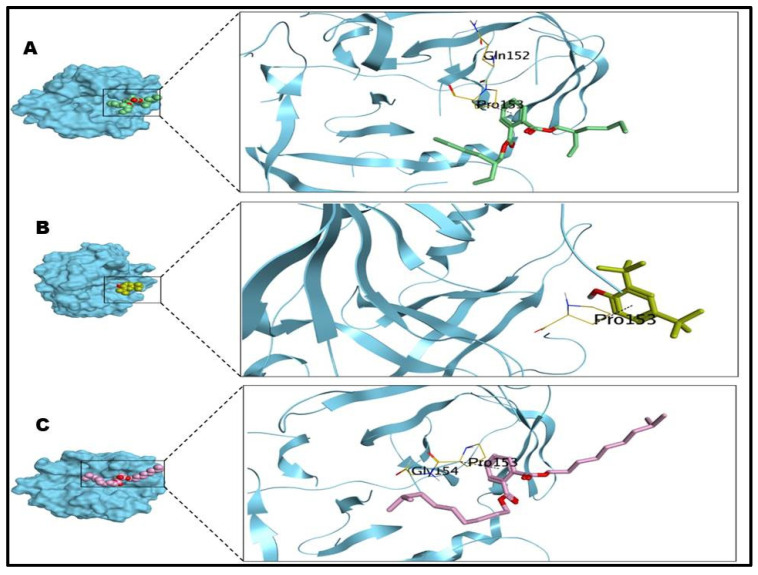
3D binding interaction and 3D protein positioning of (**A**) compound 1, (**B**) compound 2, and (**C**) compound 3 with *S. aureus* protease, with PDB entry 4ink.

**Figure 6 antibiotics-12-00464-f006:**
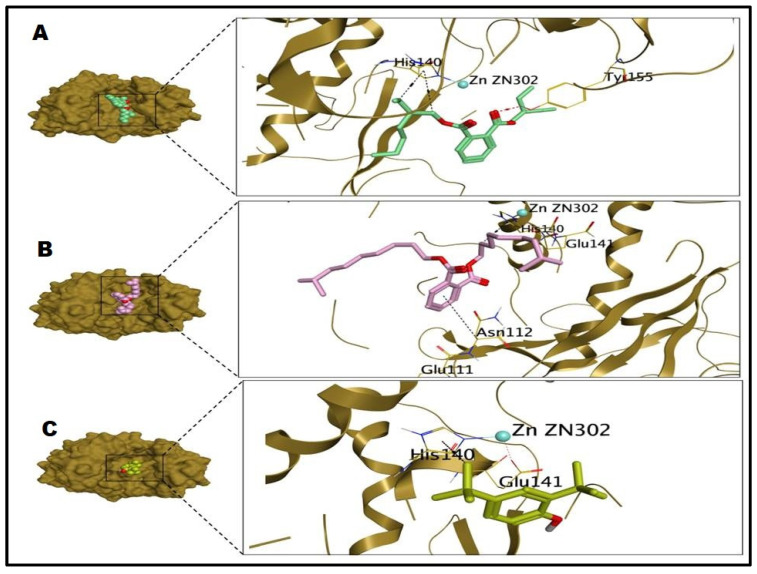
3D binding interaction and 3D protein positioning of (**A**) compound 1, (**B**) compound 2, and (**C**) compound 3 at *P. aeruginosa* elastase with PDB entry: 1ezm.

**Table 1 antibiotics-12-00464-t001:** Binding scores, RMSD values, and amino acid interactions of the investigated compounds with the *S. aureus* protease and *P. aeruginosa* elastase.

Tested Microbes	Comp. No	S Score (Kcal/mol)	RMSD (Å)	Interactions	Distances (Å)
*S. aureus*	1	−6.52	1.74	PRO153/pi-H	3.83
	2	−4.46	1.52	PRO153/pi-H	3.89
	3	−7.36	1.58	PRO153/pi-H	3.77
*P. aeruginosa*	1	−7.47	2.38	TYR155/H-acceptor GLU141/Metal HIS140/H-pi HIS140/H-pi	2.85 1.72 4.40 3.70
	2	−4.74	0.98	GLU141/Metal HIS140/H-pi	1.72 4.25
	3	−7.45	1.90	GLU141/Metal HIS140/H-pi ASN112/pi-H	1.72 4.43 4.79

## Data Availability

All data generated or analyzed during this study are included in the published article, and there are no supplementary information files.

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
