# Peer review of "Tamarindus indica Extract as a Promising Antimicrobial and Antivirulence Therapy"

_antibiotics, 2023, doi:10.3390/antibiotics12030464_

Round 1

Reviewer 1 Report

The authors presented a paper on "Tamarindus Indica Extract as a Promising Antimicrobial and Anti-virulence Therapy".

The topic is interesting and well within the aims and scopes of the Journal. However, the manuscript needs minor modifications for further publication in this journal. Here are some points which the author should consider before submitting the manuscript.

1) Title: Tamarindus indica should be italic. The species author should also be added.

2) The Abstract needs to be improved. Report the values of MIC and MBC here.

3) In Introduction: Need more introduction about Tamarindus indica.

4) Line 90: T. indica should be italic.

5) Need to improve the quality of Figures.

6) Line 181: ethanolic should not be italic.

7) Line 196: T. indica should be italic.

8) Try to improve your discussion. Need to further discussion of the results of this study with previous studies.

9) In Materials and Methods: Where and when was Tamarindus indica collected?

10) Line 469: S. aureus and P. aeruginosa should be italic.

Author Response

Many thanks for the Reviewers' comments and the opportunity to further revise the paper. We would like to thank the reviewers for their raised and thorough comments. The corrections requested by the reviewers have been done point by point as shown in the revision form. Hopefully, our revised manuscript meets the expectations of you and the reviewers and be considered for publication in antibiotic journals

Reviewer 1

The authors presented a paper on "Tamarindus Indica Extract as a Promising Antimicrobial and Anti-virulence Therapy".

The topic is interesting and well within the aims and scopes of the Journal.

Thank you for your positive comment

However, the manuscript needs minor modifications for further publication in this journal. Here are some points which the author should consider before submitting the manuscript.

We will improve the manuscript in the line of the reviewer suggestions

1) Title: Tamarindus indica should be italic. The species author should also be added.

Thank you for your excellent reviewing, it was italicized and revised

2) The Abstract needs to be improved. Report the values of MIC and MBC here.

Very thanks for your comment, both MIC and MBC values were added in the abstract

3) In Introduction: Need more introductions about Tamarindus indica.

Thank you for your comment, more information about Tamarindus indica were added

4) Line 90: T. indica should be italic.

Thank you for good inspection of our manuscript the T. indica was italicized

5) Need to improve the quality of Figures.

Thank you for your comment, all the resolution and the quality of our figures were increased and improved

6) Line 181: ethanolic should not be italic.

Thank you for your comment, It was corrected

7) Line 196: T. indica should be italic.

Thank you for your comment, It was italicized

8) Try to improve your discussion. Need to further discussion of the results of this study with previous studies.

Thank you for your comment, more information were added and it was further improved

9) In Materials and Methods: Where and when was Tamarindus indica collected?

Thank you for your comment, we added more information regarding the sources

10) Line 469: S. aureus and P. aeruginosa should be italic.

Thank you for your comment, it was italicized

Reviewer 2 Report

   Comments to the Authors

        The manuscript describes the antibacterial and anti-virulence activities of the ethanolic extract from fruits of T. indica against multi-drug resistant (MDR) bacterial strains such as Staphylococcus aureus (S. aureus) and Pseudomonas aeruginosa (P. aeruginosa) isolated from Egyptian hospital environments. The novelty of this investigation is the determination of antimicrobial mechanisms by various methods in addition to the anti-virulence activity of T. indica extract by molecular docking.

       This paper recommended to be published in current journal due to results were well fitted, described, reviewed wisely through literature, and proved its data correctness. There are few grammatical mistakes and mistypes needed to be corrected and revised carefully.

The authors are recommended to address the minor concerns listed below.

-          Line 100 “(P. aeruginosa).” should be “(P. aeruginosa)”

-          Lines 104, 110, the words gram +ve and gram –ve were written with small and in lines 119, 123, 316 with capital Gram+ve and Gram-ve letters? You should write everywhere similar, small…

-          Line 181 “ethanolic” and line 293 “extract” should not be italic

-          Lines 190, 193… “MWt” should be “MW”

-          Lines 196, 283 “T. indica” and line 290 “in vitro” should be italic

-          Compounds numbers in lines 202-220 should be bold

-          Lines 239 and 261 “Compoiund” should be “Compound”

-          Line 383 “T. indica” should be “T. indica

Author Response

Many thanks for the Reviewers' comments and the opportunity to further revise the paper. We would like to thank the reviewers for their raised and thorough comments. The corrections requested by the reviewers have been done point by point as shown in the revision form. Hopefully, our revised manuscript meets the expectations of you and the reviewers and be considered for publication in antibiotic journals

Reviewer 2

The manuscript describes the antibacterial and anti-virulence activities of the ethanolic extract from fruits of T. indica against multi-drug resistant (MDR) bacterial strains such as Staphylococcus aureus (S. aureus) and Pseudomonas aeruginosa (P. aeruginosa) isolated from Egyptian hospital environments. The novelty of this investigation is the determination of antimicrobial mechanisms by various methods in addition to the anti-virulence activity of T. indica extract by molecular docking.

This paper recommended to be published in current journal due to results were well fitted, described, reviewed wisely through literature, and proved its data correctness. 

Thank you for your positive comment

There are few grammatical mistakes and mistypes needed to be corrected and revised carefully.

The manuscript was well revised and all grammatical mistakes and mistypes were corrected

The authors are recommended to address the minor concerns listed below.

-Line 100 “(P. aeruginosa).” should be “(P. aeruginosa)”

Thank you for your comment, it was revised

-Lines 104, 110, the words gram +ve and gram –ve were written with small and in lines 119, 123, 316 with capital Gram+ve and Gram-ve letters? You should write everywhere similar, small…

Thank you for your comment, it was uniformed

-Line 181 “ethanolic” and line 293 “extract” should not be italic

Thank you for your comment, It was corrected

-Lines 190, 193… “MWt” should be “MW”

Thank you for your comment, It was modified

-Lines 196, 283 “T. indica” and line 290 “in vitro” should be italic

Thank you for your comment, they were italicized

-Compounds numbers in lines 202-220 should be bold

Thank you for your comment, they were modified

-Lines 239 and 261 “Compoiund” should be “Compound”

Thank you for your comment, It was corrected

-Line 383 “T. indica” should be “T. indica

Thank you for your comment, it was italicized